# Effect of voicing and articulation manner on aerosol particle emission during human speech

Sima Asadi[1], Anthony S. Wexler[2,3,4,5], Christopher D. Cappa[4], Santiago Barreda[6], Nicole M. Bouvier[7,8], William D. Ristenpart[1]*

1 Dept. of Chemical Engineering, University of California Davis, Davis, California, United States of America, 2 Dept. of Mechanical and Aerospace Engineering, University of California Davis, Davis, California, United States of America, 3 Air Quality Research Center, University of California Davis, Davis, California, United States of America, 4 Dept. of Civil and Environmental Engineering, University of California Davis, Davis, California, United States of America, 5 Dept. of Land, Air and Water Resources, University of California Davis, Davis, California, United States of America, 6 Dept. of Linguistics, University of California Davis, Davis, California, United States of America, 7 Div. of Infectious Diseases, Dept. of Medicine, Icahn School of Medicine at Mount Sinai, New York, New York, United States of America, 8 Dept. of Microbiology, Icahn School of Medicine at Mount Sinai, New York, New York, United States of America

* wdristenpart@ucdavis.edu

**Data Availability Statement:** All relevant data are within the manuscript and its Supporting Information files.

## Abstract

Previously, we demonstrated a strong correlation between the amplitude of human speech and the emission rate of micron-scale expiratory aerosol particles, which are believed to play a role in respiratory disease transmission. To further those findings, here we systematically investigate the effect of different 'phones' (the basic sound units of speech) on the emission of particles from the human respiratory tract during speech. We measured the respiratory particle emission rates of 56 healthy human volunteers voicing specific phones, both in isolation and in the context of a standard spoken text. We found that certain phones are associated with significantly higher particle production; for example, the vowel /i/ ("need," "sea") produces more particles than /ɑ/ ("saw," "hot") or /u/ ("blue," "mood"), while disyllabic words including voiced plosive consonants (e.g., /d/, /b/, /g/) yield more particles than words with voiceless fricatives (e.g., /s/, /h/, /f/). These trends for discrete phones and words were corroborated by the time-resolved particle emission rates as volunteers read aloud from a standard text passage that incorporates a broad range of the phones present in spoken English. Our measurements showed that particle emission rates were positively correlated with the vowel content of a phrase; conversely, particle emission decreased during phrases with a high fraction of voiceless fricatives. Our particle emission data is broadly consistent with prior measurements of the egressive airflow rate associated with the vocalization of various phones that differ in voicing and articulation. These results suggest that airborne transmission of respiratory pathogens via speech aerosol particles could be modulated by specific phonetic characteristics of the language spoken by a given human population, along with other, more frequently considered epidemiological variables.

**Funding:** WDR, ASW, and NMB received funding from the National Institute of Allergy and Infectious Diseases of the National Institutes of Health (NIAID/NIH, https://www.niaid.nih.gov), grant R01 AI110703. ASW received funding from the National Institute of Environmental Health Sciences of the National Institutes of Health (NIEHS/NIH, https://www.niehs.nih.gov) for the UC Davis Core Center, grant P30-ES023513. The funders had no role in study design, data collection and analysis, decision to publish, or preparation of the manuscript.

**Competing interests:** The authors have declared that no competing interests exist.

# Introduction

Airborne disease transmission between humans is believed to result from the release of infectious microorganisms into the air from an infected individual via expiratory activities (e.g. sneezing, coughing, talking, and breathing), and their subsequent inhalation by a nearby susceptible person [1,2,3,4,5,6,7]. Despite much research over the past few decades, it remains unclear which expiratory activities contribute most heavily to airborne disease transmission [4]. Although sneezing and coughing are commonly implicated in respiratory disease transmission, owing to the readily observable, high-velocity droplet sprays that they produce, recent studies suggest that breathing and talking may generate even larger quantities of infectious aerosol particles over time [8,9,10]. For example, Lindsley et al. [8] found that viable influenza A virus was more often detectable in cough-generated than breathing-generated particles, but because coughing occurs with much lower frequency, breathing actually may release more infectious material over time. Likewise, Yan et al. [9] detected infectious influenza virus in 39% of fine aerosol samples collected from human volunteers with influenza during 30 minutes of normal tidal breathing and occasional prompted speech. Their data further confirmed that sneezing and coughing are not necessary for the aerosolization of viable influenza virus [9].

Respiratory aerosol particles are believed to be generated primarily in the lungs during inhalation, via a "fluid film burst" mechanism in which aerosol particles are produced as a result of the clearance of fluid closures formed in the bronchioles or pores of Kohn following exhalation [11,12,13]. Specifically, as the airways collapse during exhalation, films of respiratory fluid across small-airway lumens are formed that then rupture upon the airway reopening during the following inhalation; droplets formed during the film rupture events are then carried out with the exhaled breath. Similarly, laryngeal particle generation is also believed to occur during speaking because of fluid films bursting when the vocal folds adduct and vibrate within the larynx, or during coughing and sneezing due to shear stress in the mucus-air interface within the trachea [6]. Morawska et al. [14] reported that the average particle number concentration for continuous vocalization is higher than breathing (1.1 cm$^{-3}$ for speaking and 0.1 cm$^{-3}$ for breathing), a finding they interpreted in terms of the additional contribution of laryngeal particle generation that does not occur during normal breathing. Intriguingly, we found particle emission rate during human speech is positively correlated with the voice amplitude, ranging from 1 to 50 particles per second (0.06 to 3 particles per cm$^3$) for low to high voice amplitudes [10]. Even quietly reading aloud a passage of text released significantly higher numbers of particles than different types of breathing. Furthermore, we also established the existence of "speech superemitters" who release an order of magnitude more particles than their peers [10], similar to breath "high producers" reported previously by Edwards et al. [15].

These observations clearly demonstrate the significant role of speech in aerosol particle release, but to date little attention has been paid to different types of speech. Although vocalization clearly causes emission of aerosol particles, it is unknown if some sounds release more particles than others. Previous measurements of speech aerosol particles have mainly focused on general comparisons of speaking to either breathing or coughing [16,17,18,19], or on the overall amplitude of speech [10], and few specific comparisons of different 'phones' (the basic units of speech) have been performed. One early study using high-speed photography measured the number of large droplets (larger than 5 to 10 μm in diameter) released during talking and demonstrated that consonants such as 'p', 't', 's', and 'f' produce the greatest number of droplets compared to the very few such droplets expelled by vowels [20]. This technique, however, was not capable of detecting smaller particles (smaller than 5 μm in diameter), which have subsequently been demonstrated to represent the vast majority of particles emitted by

speech [10,14]. Moreover, measurements of airflow rate during vocalization indicated that certain letters like 't' are associated with higher peak flow rate, consistent with the observed higher peak flow rates for vocalizing words like 'two' and 'ten' [21]. Our group previously measured the particle emission rate for 30 bilingual participants, reading aloud two translations of the same text, once in English and once in either Spanish, Mandarin, or Arabic [10]. We found no statistically significant difference in overall aerosol particle emissions for any of the four languages; however, the distribution of phones in the translated texts was neither measured nor controlled, so it remains unknown if specific elements of speech, which may be more common in one language than another, are associated with increased particle emission.

In this work, we directly test how vocalization of specific phones affects the particle emission rate during speech. We first investigated vowels, comparing particle emission rates for both single vowel articulations and for a series of monosyllabic words with different vowels but identical consonants. We then investigated consonants, measuring particle emission during vocalization of a series of disyllabic words containing the same repeated vowel but different consonants. Finally, we performed a detailed phonetic analysis of a passage of text, the 'Rainbow' passage widely used in linguistics research [22], to measure the dynamics of particle emission during "normal speech" and to correlate the dynamics with the underlying phonetic structure of the text. As described below, we find that certain phones consistently and significantly yield different amounts of aerosol particles during speech. We finish by discussing the implications for respiratory disease transmission via speech.

## Materials and methods

### Human subjects

We recruited 56 healthy volunteers (31 males and 25 females, ranging in age from 18 to 45 years old) by posting flyers at the University of California Davis campus over the time period May 2016 to April 2019. The University of California Davis Institutional Review Board approved this study, and all research was performed in accordance with relevant guidelines and regulations of the Institutional Review Board. Written informed consent was obtained from all participants prior to study participation and all participants completed a brief questionnaire including their age, gender, weight, height, general health status, and smoking history. Only participants who self-reported as healthy non-smokers were included in the study.

### Experimental set-up

The experimental methodology was similar to previous work [10]. In brief, the experimental setup includes an aerodynamic particle sizer (APS, TSI model 3221) placed inside a HEPA-filtered laminar flow hood that provides class 10 air to minimize background particle concentrations (Fig 1A). The APS recorded both the number and size of particles between 0.5 and 20 μm, during one-second increments. The APS also recorded the number of particles detected between 0.37 and 0.5 μm, but due to limits in resolution does not report their size more precisely. Study participants spoke into a funnel connected to the APS inlet via a conductive silicon tube; a microphone and decibel meter was placed immediately on either side of the funnel to record the vocalizations (Fig 1A). The participants were positioned such that their mouth was 1 cm from the funnel entrance. Nasal exhalations were diverted away from the APS by means of a "nose rest" that positioned the participant's nostrils outside of the funnel itself. The APS pulls air in 5 L/min, from which by design only 1 L/min of this air is sampled and the rest is filtered by the APS to be used as the sheath flow. The APS acquisition time was set to 1 second, i.e., it reported a cumulative number of particles detected per second. Note that the APS measurement module was positioned approximately 15 cm away from the participant's

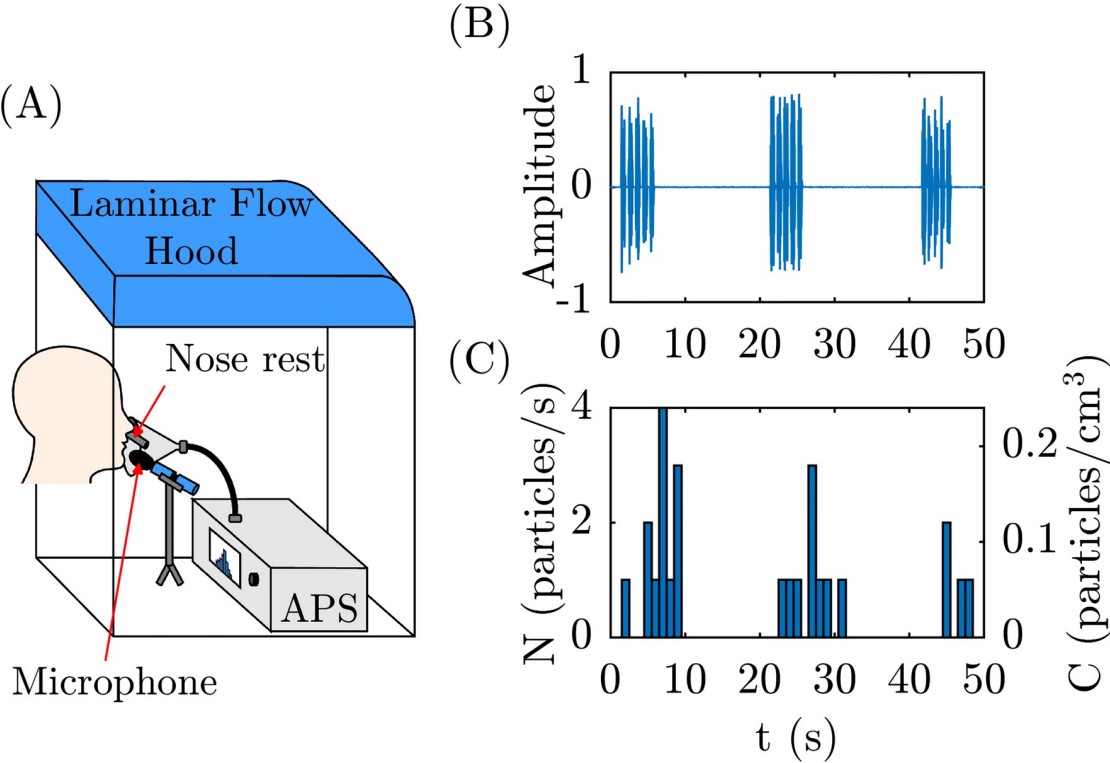

**Fig 1. Schematic of experimental setup and representative raw data.** (A) Schematic of experimental setup showing a participant talking into a funnel connected to the APS. (B) The microphone records the amplitude (arb. units) versus time for a participant saying 'papa' five times in rapid succession, followed by a 15 second pause, and repeated 3 times. (C) The APS simultaneously measures the time-resolved particle emission rate, N.

mouth, so all particles measured here had fully dried into droplet nuclei prior to measurement; if desired different correction factors available in the literature can be used to estimate the initial diameter of the particles [23]. In the context of airborne disease transmission, however, what matters is the size of the droplets upon inhalation by susceptible individual. i.e., in their fully dried state. Also note that the funnel cone was a semi-confined environment, rather than completely confined, so not all expired particles were necessarily sampled by the APS. We emphasize therefore that our measurements do not report the absolute number of particles emitted by each participant, but that this approach does allow for relative comparisons of particles emitted during vocalization of different phones.

### Experiments

**Vowel experiments.**   Participants (n = 10, 6 males, M1 to M6, and 4 females, F1 to F4) voiced /ɑ/ (the vowel sound in 'saw') for 5 seconds, followed by 15 seconds of nose breathing, repeated 3 times in succession. The participants repeated the series of three /ɑ/ vocalizations, to the best of their ability, at the same amplitude. Each participant completed 4 sets of /ɑ/ experiments, each set performed at different, self-regulated voice amplitude. Timed prompts with directions for the requested vocalization appeared on the computer screen placed in front of the participant, which displayed a timer and an amplitude (loudness) gauge to help the participants regulate their voice amplitude. The requested amplitudes were presented to participants in a random order. All participants repeated the same experiment with vowels /i/ (the vowel sound in 'need') and /u/ (the vowel sound in 'mood').

We denote $N_V$ as the particle emission rate for repeating each vowel 3 times, where the subscript 'V' represents 'vowels.' Note that particle emission rate differs from person to person, especially due to the existence of subjects known as superemitters, who emit an order of magnitude greater number of particles than other individuals (see for example participant F4 in reference [10]). To address this disparity and allow comparison between individuals, we normalized $N_V$ by $N_{V,avg}$, which is defined as the average particle emission rate for each individual over four different amplitudes tested.

In our previous work we showed that there is positive correlation between particle emission rate and root mean square voice amplitude, either when saying /ɑ/ for several seconds continuously or for reading a passage of text aloud [10]. Since all participants tend to vocalize at different amplitudes, it was therefore important to develop a calibration curve for particle emission rate versus amplitude for each individual. Accordingly, in the vowel experiments, participants first repeated a specific vowel (/ɑ/ as in 'saw', /i/ as in 'need', or /u/ as in 'mood') at four different self-regulated amplitudes, and the number of emitted particles and voice amplitude were simultaneously recorded. To facilitate comparisons of particle emission rates corresponding to vocalization of specific phones, we chose a 'standard amplitude' of $A_{rms} = 0.1$, equalling approximately 85 decibels measured 6.5 cm away from the participant's mouth over a background noise of roughly 65 decibels. Qualitatively, this amplitude corresponds to a comfortable speaking voice. To perform comparisons we used a power law regression of the normalized particle emission rate versus root mean square amplitude for each individual to calculate the interpolated particle emission rate at $A_{rms} = 0.1$, denoted here as $\tilde{N}_V/\tilde{N}_{V,avg}$ (where the tilde denotes the interpolated value).

**Monosyllabic word experiments.** Participants (n = 10, 6 males, and 4 females) repeated 12 monosyllabic words in the form of 'h.vowel.d': 'had', 'head', 'who´d', 'hud', 'heard', 'hood', 'hawed', 'heed', 'hayed', 'hod', 'hoed', and 'hid', with International Phonetic Alphabet (IPA) notation of /hæd/, /hɛd/, /hud/, /hʌd/, /hɜrd/, /hʊd/, /hɔd/, /hid/, /heid/, /hɑd/, /hoʊd/, and /hɪd/ respectively. Note that some of these words are very similar in pronunciation; in particular, 'hod' and 'hawed' have been merged in California and most of North America for decades, but they are included here for completeness and to serve as internal control. Using an intermediate voice amplitude (i.e., a normal conversational voice), the participants repeated a particular word 5 times in 5 seconds, followed by 15 seconds of nose breathing, repeating the sequence 3 times; in this manner, each word was repeated 15 times by each participant.

Since participants were asked to repeat each word at their normal conversational voice amplitude, small variations unavoidably occurred in the average amplitude between participants. To compare the particle emission rate of different words at the same amplitude, we used the power law regression exponent for each specific individual measured while they said /ɑ/ (cf. Fig 2A) to interpolate the particle emission rate of each monosyllabic word to the standard voice amplitude $A_{rms} = 0.1$. That quantity is denoted as $\tilde{N}_M$, where the subscript "M" denotes "monosyllabic" and the tilde denotes the interpolated value at $A_{rms} = 0.1$. Again to address the challenge of superemitters, we normalized $\tilde{N}_M$ with its average for each individual over all 12 different monosyllabic words each repeated 15 times, indicated as $\tilde{N}_{M,avg}$.

**Disyllabic word experiments.** Participants (n = 30, 16 males, and 14 females) repeated 14 disyllabic words, each composed of a distinct repeated phone and vowel /ɑ/: 'baba', 'papa', 'dada', 'tata', 'gaga', 'kaka', 'zaza', 'sasa', 'vava', 'fafa', 'haha', 'shasha', 'nana', and 'mama'. The chosen consonants include *voiced plosives* (/b/, /d/, and /g/), *voiceless plosives* (/p/, /t/, and /k/), *voiced fricatives* (/z/, and /v/), *voiceless fricatives* (/s/, /f/, /h/, and /ʃ/), and *nasals* (/m/, and /n/). Similar to the monosyllabic word experiments, the participants repeated a particular word 5 times in 5 seconds, followed by 15 seconds of nose breathing at an intermediate loudness.

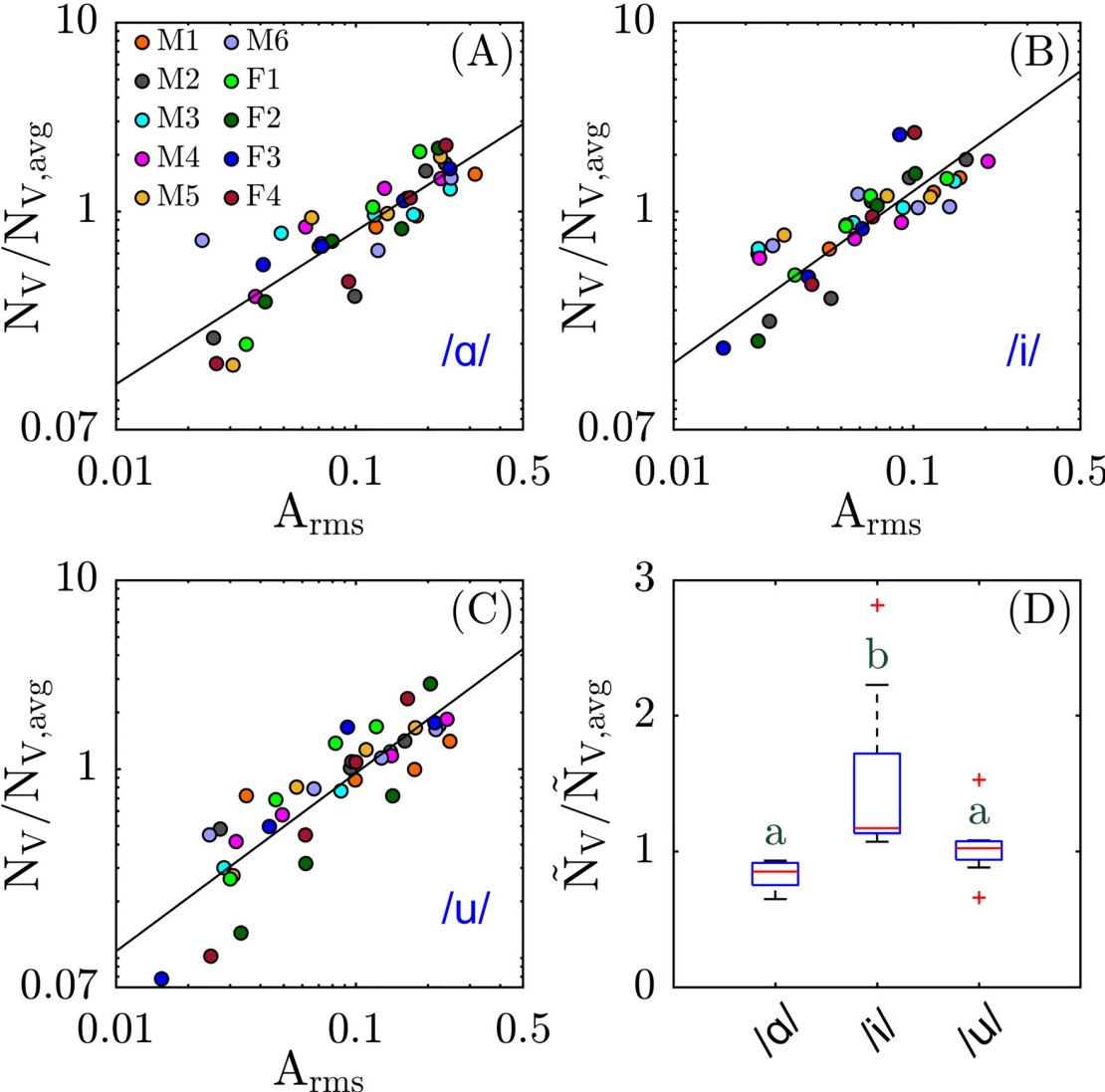

**Fig 2. Particle emission rate of vowels.** Normalized particle emission rate, $N_V/N_{V,avg}$, versus root mean square amplitude, $A_{rms}$, while saying **(A)** /ɑ/ (the vowel sound in 'saw'), **(B)** /i/ (the vowel sound in 'need'), and **(C)** /u/ (the vowel sound in 'mood') for 4 different amplitudes by 10 participants, 6 males (denoted as M1 to M6), and 4 females (denoted as F1 to F4). Solid lines are power law fits with exponent **(A)** 0.81, **(B)** 0.91, and **(C)** 0.94, correlation coefficient **(A)** 0.86, **(B)** 0.74, and **(C)** 0.82, and Pearson's p value **(A)** $2.1 \times 10^{-12}$, **(B)** $5.9 \times 10^{-8}$, and **(C)** $6.7 \times 10^{-11}$. **(D)** Boxplot of normalized particle emission rate calculated at $A_{rms} = 0.1$, $\tilde{N}_V/\tilde{N}_{V,avg}$, (sample size n = 10). Scheffe groups are indicated with letters; groups with no common letter are considered significantly different (p < 0.05).

Using an intermediate voice amplitude, 30 participants (16 males, and 14 females) performed the disyllabic word experiments. Similar to the monosyllabic word experiments, the particle emission rates were interpolated to $A_{rms} = 0.1$ using the power law regression exponent obtained from saying /ɑ/ for each individual participant (cf. Asadi et al. [10] supplementary Fig. S5 for /ɑ/ data). Then the particle emission rate of disyllabic words at $A_{rms} = 0.1$, denoted as $\tilde{N}_D$ where the subscript 'D' denotes disyllabic, was normalized using the average for each individual over 14 different disyllabic words each repeated 15 times, represented as $\tilde{N}_{D,avg}$.

**Rainbow passage experiments.** Participants (n = 18, 10 males, M7 to M16, and 8 females, F5 to F12) read aloud a 330-word excerpt of text in English, known in linguistics research as the Rainbow passage [22]. Participants were asked to read the Rainbow passage aloud using an intermediate level of loudness and comfortable pace, as a "normal conversational voice." The APS recorded the particle emission rate, and the microphone recorded amplitude versus time.

To analyze the resulting data, first we conceptually divided the Rainbow passage into 46 phrases (S1 Table), each composed of 10 syllables on average, such that each phrase had different duration but roughly equal numbers of phones. Next, based on the IPA notation of each phrase, we counted how many phones in each phrase fall into the following 8 phonetic categories: vowels, voiced plosives, voiceless plosives, voiced fricatives, voiceless fricative, nasals, approximants, and trills. Finally, we calculated the particle emission rate that occurred during vocalization of each phrase, denoted as $N_R$ where the subscript 'R' denotes 'Rainbow,' and defined as the number of particles emitted while a specific phrase was read aloud, divided by the duration in seconds of that specific phrase. Similar to the previous measures, $N_R$ was normalized by the average particle emission rate $N_{R,avg}$ of each specific individual over all 46 phrases, to prevent superemitters from skewing the results.

## Statistical analysis

Data analysis was performed in MATLAB (MathWorks), with data fits performed as noted in figure legends. Pearson's linear correlation coefficients and p values were calculated for linear fits. Box-and-whisker plots show the median (red line), interquartile range (blue box), range (black whiskers), and outliers (red plus signs), defined as values that exceed 2.7 standard deviations (providing approximately 99% coverage). Stata/IC 15 was used to perform mixed-effects linear regression to account for person-level correlations, and considering that we had only one primary random effect (between-person variability), all variances were set to be equal with zero covariances. Post hoc pairwise comparisons were performed and adjusted for multiple comparisons using Scheffe's method. Scheffe groups are indicated with green letters above each box plot; groups with no common letter are considered significantly different (p < 0.05).

We emphasize that prior to this work, there were no measurements of aerosol production versus different phones available in the literature; to our knowledge, this study is the first to investigate this question and thus should be considered exploratory in nature. In other words, we had no data available to inform a pre-study statistical power calculation. Nonetheless, we could perform a sample post hoc power calculation to compare one pair of our disyllabic word data (e.g., dada vs. papa) to assess statistical power. Stata/IC 15 was used to perform a paired test comparing two correlated means using a 5%-level two-sided test and sample size of n = 30, resulting in a statistical power of 0.89. Similar results are obtained from pairwise comparison of other disyllabic pairs (n = 30) and monosyllabic pairs (n = 10). More complicated power calculations for multi-level mixed effects models can be used; however, in this study we are more interested in establishing the existence of difference between particle emission rates of different phonemes rather than looking for clinically meaningful differences.

## Results

Representative raw data for a single participant repeating the word 'papa' (/pɑpɑ/) is shown in Fig 1. The simultaneous microphone recording (Fig 1B) and APS measurements (Fig 1C) demonstrate that when the vocalization starts, after approximately a 2 second lag required for the particles to reach the APS sensor, the observed number of particles increases. When the participant resumes nose breathing, the number of particles decreases back to zero, as expected for the background HEPA-filtered air.

Fig 2A to 2C show the normalized particle emission rate, $N_V/N_{V,avg}$, versus root mean square amplitude, $A_{rms}$, for saying /ɑ/, /i/, and /u/, respectively. Note these values center around 1 because of the normalization with each individual's average emission rate (to account for superemitters); examination of the non-normalized rates $N_V$ reveals a linear increasing trend between $N_V$ and $A_{rms}$ for each specific individual, but the aggregated data for all 10 participants are weakly correlated (S1 Fig). Fig 2A to 2C demonstrate that the normalized particle emission rate, $N_V/N_{V,avg}$, is strongly correlated with $A_{rms}$ for the three vowels tested here.

Interpolation of the rate data in Fig 2A to 2C to the standard amplitude of $A_{rms} = 0.1$ reveals a significant difference in particle emission rates (Fig 2D). The interpolated, normalized rate $\tilde{N}_V/\tilde{N}_{V,avg}$ for saying /i/ is significantly higher than for either /ɑ/ and /u/, with almost no overlap between the boxplots for /i/ versus /ɑ/ or /u/. In contrast, no significant difference is observed between /ɑ/ and /u/.

The observation that the vowel /i/ generates more particles on average was reproduced independently in the second set of experiments, in which the participants repeated 12 monosyllabic words all starting with 'h' and ending in 'd', and with a single vowel phone in between (which in English may be represented by one or more letters). The boxplot in Fig 3 presents the normalized particle emission rate at $A_{rms} = 0.1$ ($\tilde{N}_M/\tilde{N}_{M,avg}$) for n = 10 participants. According to the statistical analysis of these data, the normalized particle emission rate $\tilde{N}_M/\tilde{N}_{M,avg}$ is significantly higher for the word 'heed' (IPA notation /hid/) than for the other words. The normalized particle emission rates for the two phones, 'hayed' (/heɪd/) and 'who'd' (/hud/) were statistically intermediate between 'heed' and the rest of the monosyllabic words.

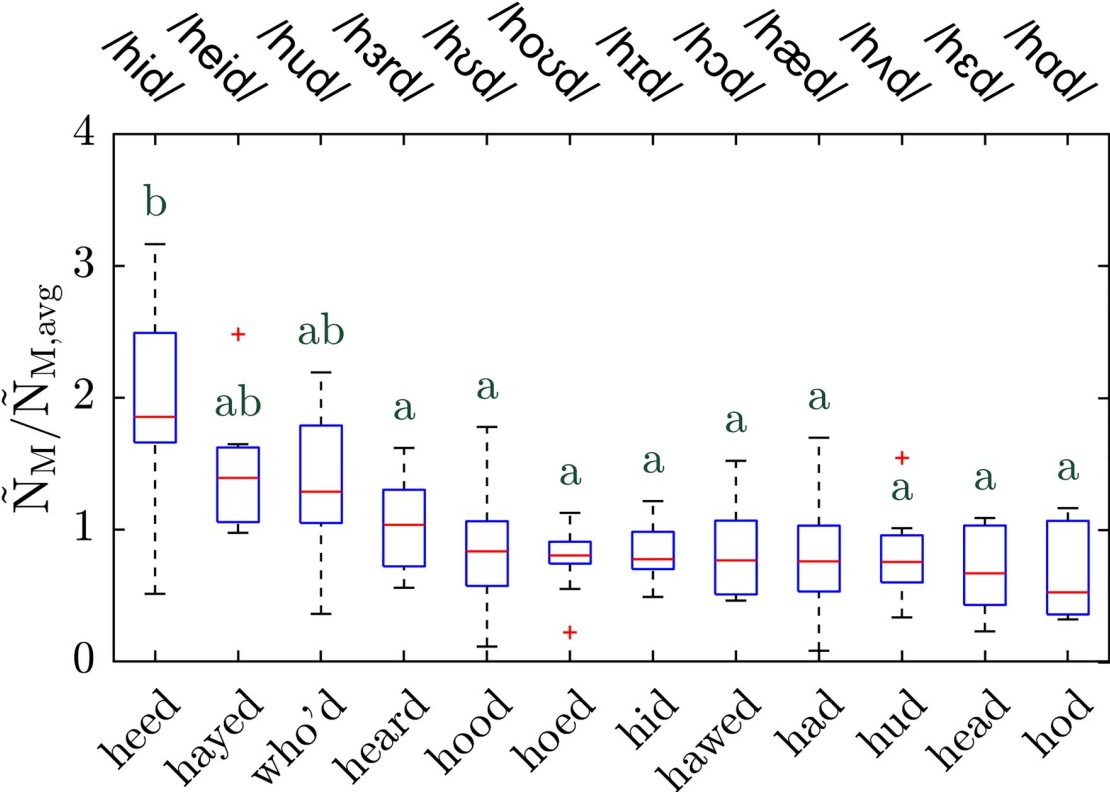

**Fig 3. Particle emission rate of monosyllabic words.** Boxplot of normalized particle emission rate while repeating 12 monosyllabic words calculated at $A_{rms} = 0.1$, $\tilde{N}_M/\tilde{N}_{M,avg}$ (sample size n = 10). Top x-axis shows the IPA notation of each word. Scheffe groups are indicated with letters; groups with no common letter are considered significantly different (p < 0.05).

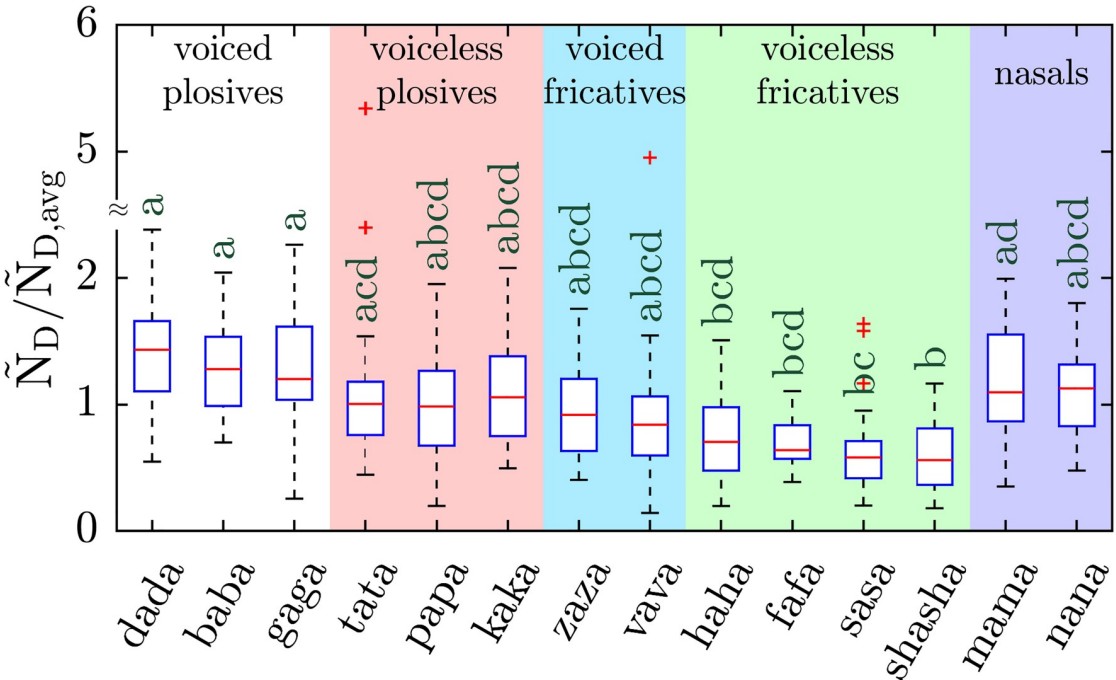

**Fig 4. Particle emission rate of disyllabic words.** Boxplot of normalized particle emission rate while repeating 14 disyllabic words calculated at $A_{rms} = 0.1$, $\tilde{N}_D/\tilde{N}_{D,avg}$, (sample size n = 30). Each background color represents the words including consonants which have similar voicing and articulation manner. Scheffe groups are indicated with letters; groups with no common letter are considered significantly different (p < 0.05).

The other nine monosyllables were statistically indistinguishable. Importantly, on average, the vowel /i/ in 'heed' yielded twice as many particles as the other monosyllables, qualitatively consistent with the results shown in Fig 2D. Non-normalized particle emission rate and the corresponding concentration for monosyllabic words are presented for reference in S2 Fig.

The first two experiments focused on vowels, alone and in simple words, so to examine the effect of different consonants on particle emission rate a third set of experiments was performed, in which participants said aloud 14 disyllabic words, each composed of a consonant and the vowel /ɑ/ repeated twice (e.g., "mama"). The chosen consonants include *plosives* (/b/, /d/, /g/, /p/, /t/, and/ k/), in which vocal tract airflow is momentarily stopped and then released; *fricatives* (/z/, /v/, /s/, /f/, /h/, and /ʃ/), in which airflow is forced through a narrow channel formed by bringing together two articulators such as tongue, teeth, palate, or lips; and *nasals* (/m/, and /n/), in which airflow through the oral cavity stops while nasal airflow continues. These phones can further be characterized as *voiced* if the vocal cords vibrate during articulation (/b/, /d/, /g/, /z/, /v/, /m/, and /n/) or *voiceless* if vocal cords do not vibrate and stay open during phonation (/p/, /t,/ /k/, /s/, /f/, /h/, /ʃ/). Fig 4 shows the boxplot of the normalized, interpolated emission rates $\tilde{N}_D/\tilde{N}_{D,avg}$ for all 14 disyllabic words tested for which the non-normalized particle emission rates and concentrations are presented in S3 Fig. Statistical analysis using Scheffe's method suggests that, in general, voiced consonants yield more aerosol particles than voiceless, and that plosives yield more particles than fricatives. Specifically, the disyllabic words including voiced plosives (/d/, /b/, and /g/) release more particles than words including voiceless fricatives (/f/, /h/, /s/, and /ʃ/). Likewise, the voiceless plosive /t/ ('tata') releases more particles than the voiceless fricative /ʃ/ ('shasha'), and 'mama', which is a nasal-vowel disyllabic combination, yields more particles than 'sasa' and 'shasha,' which repeat voiceless fricative

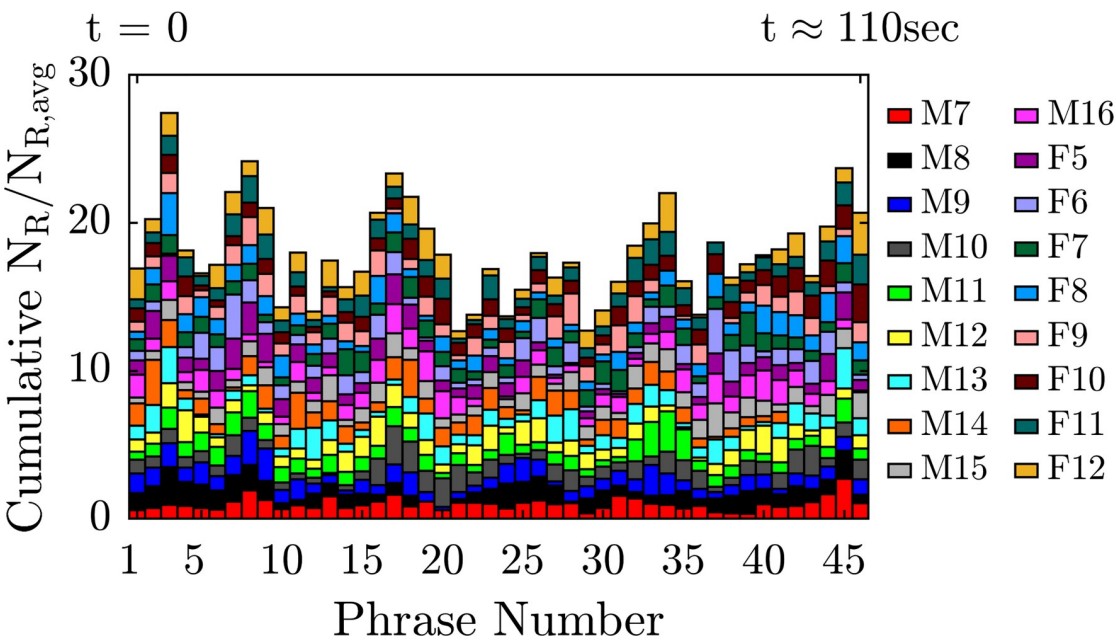

**Fig 5. Particle emission rate while reading the Rainbow passage.** Stacked bar plot of normalized particle emission rate, $N_R/N_{R,avg}$, for 18 participants (10 males, M7 to M16, and 8 females, F5 to F12) while reading Rainbow passage with an intermediate loudness, versus "phrase number" for 46 phrases in total. Each phrase was composed of approximately 10 syllables; see S1 Table for details.

consonants. In general, the voiced stops–plosives and nasals–yielded on average the most particles, and voiceless fricatives the least.

The preceding results focused on repetition of individual sounds and words, which rarely occurs in everyday speech. Our tests with a standard passage of text (the Rainbow passage) indicate that some phones do emit more particles on average even during regular speech. The stacked bar plot in Fig 5 shows the cumulative normalized particle emission rate for each phrase vocalized in the Rainbow passage. The results indicate that some phrases systematically yielded more particles than others. For example, phrase number 3, composed of the words "*The rainbow is a division of white*," yielded on average the largest number of particles, with cumulative $N_R/N_{R,avg} = 27.4$ (where cumulative indicates that the results for all 18 individuals were summed). In contrast, phrase 21, "*a sign from the gods to foretell war*," yielded the least, with cumulative $N_R/N_{R,avg} = 12.7$. Similar spikes occur when the raw data is simply plotted versus clock time rather than versus phrase number (S4 Fig), although the data are more difficult to interpret because each individual reads at a slightly different pace.

To assess for a correlation between the particle emission rate and the phonetic distribution of each phrase, we calculated the fraction of each phonetic category in each phrase (S1 Table). For example, a phrase with 10 phones, of which 5 are vowels, would have a 'vowel fraction' of 0.5. We emphasize that this fractional measure is approximate, since it neglects the duration of the vowels; it is possible for two different phrases with equivalent vowel fractions to have very different duration. Nonetheless, using this approximate measure indicates that phrases with higher vowel fractions yield significantly larger particle numbers (Fig 6A). Likewise, phrases with higher voiceless fricative fractions yielded significantly smaller numbers of particles (Fig 6B). The other measured phonetic fractions did not yield statistically significant correlations with the numbers of emitted particles (not shown).

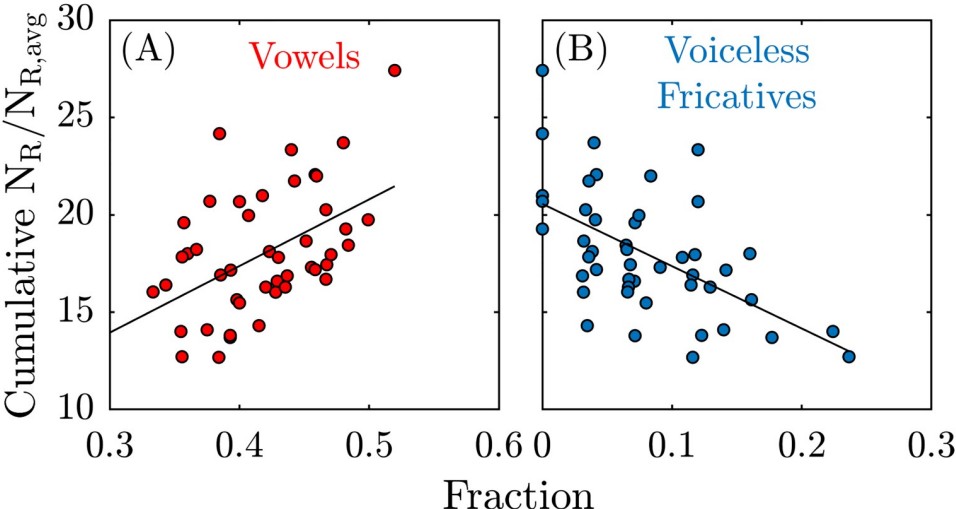

**Fig 6. Correlation between particle emission rate and phonetic fractions.** Cumulative normalized particle emission rate for 18 individuals, while reading the Rainbow passage, versus **(A)** fraction of vowels, and **(B)** fraction of voiceless fricatives in each phrase of the Rainbow passage. Solid lines are the best linear fit, with correlation coefficient **(A)** 0.48, and **(B)** -0.56, and Pearson's p value **(A)** $6.4 \times 10^{-4}$, and **(B)** $4.4 \times 10^{-5}$.

## Discussion

The results presented here clearly indicate that respiratory particle emission rate varies for different types of speech, and that the emission rate is correlated with both voicing and articulation manner of different vowels and consonants. Vowels are voiced phones produced with no obstruction of airflow, primarily classified based on the position of tongue and lips. For the vowels tested here, /i/ and /ɑ/ have the lowest and highest perceived loudness given a comparable amount of acoustic power [24]. Even if two vowels are generated exactly the same way initially in the larynx via vibration of the vocal folds, as the sound waves travel through the vocal tract their characteristics (e.g. energy) change. The vocal tract tube acts as a resonance tube with an irregular cross section with different average diameter for each vowel, causing the vowels to sound different [24]. The different acoustic powers associated with different vowels suggests that to reach a similar perceptual loudness more energy is required for vocalization of /i/ than /ɑ/ or /u/. Because the airstream from the lungs is the main source of energy for speech [25], and because the exhaled airstream carries aerosol particles [11], the larger energy requirement for the articulation of /i/ is presumably at least partially responsible for its higher particle emission rate.

Unlike the vowels, consonants are classified based on the manner of both articulation and voicing. Articulators such as the lips, teeth, hard palate, and tongue shape and control airflow through the oral cavity. As mentioned earlier, plosives are generated via a complete stop of airflow and its sudden release (similar to a puff); a nasal obstruent, when followed by a vowel, creates a similar airflow pattern. Conversely, fricatives involve a continuous weak airflow through a small opening, created by bringing two articulators (such as teeth or lips) together. In fact, prior observations have established that certain consonants, such as the plosive 't', are associated with larger egressive pulmonic airstreams compared to the fricative 'f' [21, 26]. Therefore, the difference between airflow patterns for different consonants may explain the higher particle emission rates observed for plosives and nasals in comparison to fricatives. Furthermore, consonants with similar articulation manner can be different in terms of voicing. Vocal fold vibration is hypothesized to be an important mechanism of aerosol particle generation [6],

which potentially helps to explain the higher particle emission rates for voiced consonants observed here.

The results also suggest that vowels release more aerosol particles than voiceless fricatives. One possible reason is that vowels are known in general to be louder than consonants and, as we showed in our previous work, louder speech releases more aerosol particles [10]. Furthermore, there is no obstruction in the vocal tract when producing vowels, so there is no barrier to airflow that carries particles out. Finally, the laryngeal particle generation mode present for voiced phonemes such as vowels [14] does not contribute to the particle generation rate of voiceless fricatives.

There are important epidemiological implications for respiratory disease transmission if certain phones release more aerosol particles. Different languages have different distributions of phones [27], as well as different average rates of speech [28], suggesting that airborne transmission of respiratory pathogens could be modulated by the phonetic characteristics of the language spoken. Indeed, at least one researcher has speculated that airborne disease transmission could be modulated by language spoken [29]. In the 2003 SARS epidemic, many American visitors to China became infected while allegedly zero Japanese visitors became infected, despite the fact that there were many more visitors from Japan during that same time period. Although there is controversy about the accuracy of the reported number of cases [30], the apparent discrepancy led Inouye to point out that the Japanese language lacks many aspirated phones, while English has many, and to speculate that this difference might increase the likelihood of transmission from infected Chinese speaking English versus Japanese to visitors [29]. In a follow-up study they analysed the strength of puffs generated during reading a passage of text in English, Japanese, and Chinese [31]. They recorded the wind pressure of the airflow near the mouth as a measure for initial velocity of airflow and found that for both bilingual and monolingual subjects the wind pressure and puff strength are weaker for Japanese language than English and Chinese; in contrast, they found no significant difference between English and Chinese languages. The results presented here clearly demonstrate that aerosol particle emission rates (rather than just the eggresive flow rate) vary systematically with different speech articulations. As a result, many differences between languages, such as the expected frequency of consonants versus vowels in a given utterance, may modulate airborne disease transmission.

## Conclusions

In summary, we compared particle emission rates for different types of speech including individual vowels, monosyllabic words, and disyllabic words. Our results confirm that certain vowels such as /i/ and consonants such as /d/ have higher particle emission rates than others. Likewise, particle emission rates during phonetically different phrases in a passage of text showed higher emission rates for phrases with higher fraction of vowels and lower fractions of voiceless fricatives. We interpret our observations in terms of egressive airflow rate of different phones, which is known to vary significantly with voicing and articulation manner of each phone. Considering the different distribution of phones between different languages, the results presented here lend credence to the hypothesis that individual vocalization patterns, including language spoken, could be important epidemiological metrics, and as such merit closer attention. We note that our previous work [10] examining expiratory particle emission while reading a passage of text (Chapter 24 of "The Little Prince") in different languages–Spanish, Mandarin, and Arabic–yielded data that could be analyzed in more detail by linguists with expertise in those languages. Moreover, it has been previously reported that the number of cough aerosols released during illness was significantly larger compared to subsequent post-

illness measurements [32,33]. This result raises the question of whether a similar trend would be observed for talking and breathing when comparing healthy and ill subjects. Other cultural practices (such as average distance between speakers during conversation, or typical speaking rate or loudness) could likewise modulate respiratory disease transmission. Our results here serve as a proof of concept to motivate further research efforts with larger sample sizes and other languages.

## Supporting information

**S1 Table. Details of phonetic analysis for Rainbow passage.** Blue and green columns show the number and fraction of each phoneme category in phrase, respectively.
(PDF)

**S1 Fig. Particle emission rate/concentration of vowels.** Particle emission rate ($N_V$)/concentration ($C_V$) versus root mean square amplitude, $A_{rms}$, while saying (A) /ɑ/ (the vowel sound in 'saw'), (B) /i/ (the vowel sound in 'need'), and (C) /u/ (the vowel sound in 'mood') for 4 different amplitudes by 10 participants, 6 males (denoted as M1 to M6), and 4 females (denoted as F1 to F4). Solid lines are power law fits with exponent (A) 0.80, (B) 0.70, and (C) 0.91, correlation coefficient (A) 0.54, (B) 0.14, and (C) 0.34, and Pearson's p value (A) $2.8 \times 10^{-4}$, (B) 0.37, and (C) 0.03.
(PDF)

**S2 Fig. Particle emission rate/concentration of monosyllabic words.** Boxplot of particle emission rate ($N_M$)/concentration ($C_M$) while repeating 12 monosyllabic words, $N_M$, (sample size n = 10). Top x-axis shows the IPA notation of each word.
(PDF)

**S3 Fig. Particle emission rate/concentration of disyllabic words.** Boxplot of particle emission rate ($N_D$)/concentration ($C_D$) while repeating 14 disyllabic words, $N_D$, (sample size n = 30). Each background color represents the words including consonants which have similar voicing and articulation manner.
(PDF)

**S4 Fig. Particle emission rate/concentration for reading Rainbow passage.** Bar plots of particle emission rate ($N_R$)/concentration ($C_R$) versus time for reading Rainbow passage for 18 participants (10 males denoted as M7 to M16, and 8 females denoted as F5 to F12).
(PDF)

## Acknowledgments

We thank P. Dayal for assistance with statistical analysis, and J. Liang, M. Q. Alsindi, M. I. Perez-Vargas, and R. Mavlingcar for assistance with data collection and analysis.

## Author Contributions

**Conceptualization:** William D. Ristenpart.

**Data curation:** Sima Asadi, William D. Ristenpart.

**Formal analysis:** Sima Asadi.

**Funding acquisition:** Anthony S. Wexler, Nicole M. Bouvier, William D. Ristenpart.

**Investigation:** Sima Asadi.

**Methodology:** Sima Asadi, Anthony S. Wexler, Christopher D. Cappa, Santiago Barreda, Nicole M. Bouvier.

**Project administration:** William D. Ristenpart.

**Resources:** Nicole M. Bouvier.

**Supervision:** Anthony S. Wexler, William D. Ristenpart.

**Validation:** Sima Asadi.

**Visualization:** Sima Asadi.

**Writing – original draft:** Sima Asadi.

**Writing – review & editing:** Sima Asadi, Anthony S. Wexler, Christopher D. Cappa, Santiago Barreda, Nicole M. Bouvier, William D. Ristenpart.

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
