## [Decision Letter · Decision Letter 0]

1 Oct 2019

PONE-D-19-20208

Effect of Voicing and Articulation Manner on Aerosol Particle Emission during Human Speech

PLOS ONE

Dear Prof. Ristenpart,

Thank you for submitting your manuscript to PLOS ONE. After careful consideration, we feel that it has merit but does not fully meet PLOS ONE's publication criteria as it currently stands. Therefore, we invite you to submit a revised version of the manuscript that addresses the points raised during the review process.

All three reviewers have provided very helpful comments, please provide a point-by-point response letter to the comments and questions raised. As you have previously reported the positive association between amplitude of speech and the aerosol emission rate, it would be helpful if you can comment on the level of amplitudes of different "phones" investigated in the current study. 

We would appreciate receiving your revised manuscript by Nov 15 2019 11:59PM. To enhance the reproducibility of your results, we recommend that if applicable you deposit your laboratory protocols in protocols.io, where a protocol can be assigned its own identifier (DOI) such that it can be cited independently in the future. For instructions see: http://journals.plos.org/plosone/s/submission-guidelines#loc-laboratory-protocols

We look forward to receiving your revised manuscript.

Sincerely,

Hui-Ling Yen

Academic Editor

PLOS ONE

Journal Requirements:

'We thank the National Institute of Allergy and Infectious Diseases of the National Institutes of Health

(NIAID/NIH), grant R01 AI110703, and the NIEHS UC Davis Core Centre, grant P30-ES023513, for supporting

this research.'

'WDR, ASW, and NMB received funding from the National Institute of Allergy and

Infectious Diseases of the National Institutes of Health (NIAID/NIH,

https://www.niaid.nih.gov), grant R01 AI110703. ASW received funding from the

National Institute of Environmental Health Sciences of the National Institutes of Health

(NIEHS/NIH, https://www.niehs.nih.gov) for the UC Davis Core Center, grant P30-ES023513. The funders had no role in study design, data collection and analysis, decision to publish, or preparation of the manuscript.'

Additional Editor Comments (if provided):

Reviewers' comments:

Reviewer's Responses to Questions

**Comments to the Author**

1. Is the manuscript technically sound, and do the data support the conclusions?

Reviewer #1: Yes

Reviewer #2: Yes

Reviewer #3: Yes

2. Has the statistical analysis been performed appropriately and rigorously? 

Reviewer #1: I Don't Know

Reviewer #2: Yes

Reviewer #3: Yes

3. Have the authors made all data underlying the findings in their manuscript fully available?

Reviewer #1: Yes

Reviewer #2: Yes

Reviewer #3: Yes

4. Is the manuscript presented in an intelligible fashion and written in standard English?

Reviewer #1: Yes

Reviewer #2: Yes

Reviewer #3: Yes

5. Review Comments to the Author

Reviewer #1: This manuscript reports a new set of measurements of the roles of “phones” in expired droplet generation following their early work.

Asadi S, Wexler AS, Cappa CD, Barreda S, Bouvier NM, Ristenpart WD. Aerosol emission and superemission during human speech increase with voice loudness. Scientific Reports. 2019;9:2348.

The authors now found that the “certain phones are associated with significantly higher particle production”, which is not surprising following the existing data in the literature, but with a more detailed exploration.

A total of 56 subjects were tested in total, however, for each group of tests, e.g. vowel experiments, only 14 subjects were tested. Except the disyllabic word experiments, the sample size is not large, which is a pity.

It is important to recognize that the authors only measured the droplet sizes at the site of the measurement, i.e. somewhere within APS, not at the exit. On the journey from the mouth exit, entering the funnel, flowing through the connection tube, the droplets are expected to evaporate and different sizes of droplets would experience different evaporation rates. This should probably be added in the limitation discussion.

Only a normalized particle emission rate is presented to address significant individual emission differences. It is not surprising to me that different phones produce different number of droplets. We are more interested how many are produced. The authors should provide the exact emission rates n some suitable format at least in Supplementary Information. I could not access S1 so that the information might be there (but sounds only for Rainbow passage).

The authors discussed their work in terms of the speculation by Inouye (2003) (note not an original paper, and no data was presented) and Inouye and Sugihara (2015) where some kinds of pressure was measured. Both are not a great paper. In Inouye and Sugihara (2015), the claim that Japanese language seems to produce a smaller pressure than English and Chinese. Many questions existing in their measurement if one reads their paper carefully. I have personally experienced when some Japanese friends spoke “wildly” after drink, I did not see any such “low pressure” phenomenon. I believe that the statements made in this para are not appropriate.

The work by the authors in this manuscript in Asadi et al (2019) showed that there were no statistically significant difference in overall aerosol particle emissions for English, Spanish, Mandarin and Arabic, though “distribution of phones in the translated texts was neither measured nor controlled”. Why not measure it as you have the original text?

Reviewer #2: The paper presents measurements of aerosol production during various parts of human speech. The work is unique and interesting, and provides insights into human aerosol production, which is an important factor in airborne disease transmission. The paper is well written and easy to follow, which is greatly appreciated. I have only a few minor comments.

The authors compare aerosol production during the enunciation of different vowel sounds, and during the enunciation of different consonants. Can they compare the relative aerosol production for vowels vs. consonants? Figure 6A might suggest that vowels produce more aerosols than consonants, but I may be misinterpreting the data. If the authors’ data doesn’t allow them to make this comparison, that is fine, but if the comparison is possible, then it would be interesting.

The normal breathing rate for an adult at rest is typically 5-10 liters/minute. As the authors note, their aerosol spectrometer draws air at 5 liters/minute, so they are measuring a sample of the exhaled breath, not all of the particles that are exhaled. Does the rate of exhalation vary significantly during speech? If so, this might affect the measurements, in that if a person was producing twice as many particles when pronouncing one phone as vs. another but also exhaling twice as fast, the aerosol concentration measured by the APS would be the same. On the other hand, if variations in flow due to speech are small, then the effect would be minimal.

What was the time resolution for the APS? Did it report cumulative counts for every second, or every 2 seconds, or something else?

In most places in the text, the authors use the term “voiceless”, but sometimes they use “unvoiced”. Are these terms equivalent? If so, it may be best to use just one term, since I expect that most of the readers (like me) will not be familiar with the terminology of linguistics.

If the authors plan to continue this line of research, it would be very interesting to see if a difference in aerosol emission or emission of pathogens could be detected in, for example, people with colds or influenza while they are speaking vs. just breathing.

Reviewer #3: The study aimed (n=56) to investigate the effect of different ‘phones’ (the basic sound units of speech) on the emission of particles from the human respiratory tract during speech. Certain phones were associated with significantly higher particle production. In phrases, a positive correlation was observed between the vowel content and particle production.

Minor revisions:

1 -Indicate the underlying covariance structure used in the mixed effects linear model and the criteria for selecting it.

2-State and justify the study’s target sample size with a pre-study statistical power calculation. The power calculation should include: sample size, alpha level (indicating one or two-sided), minimal detectable difference and statistical testing method.

6. PLOS authors have the option to publish the peer review history of their article (what does this mean?). If published, this will include your full peer review and any attached files.

Reviewer #1: No

Reviewer #2: No

Reviewer #3: No

---

## [Author Response · Author response to Decision Letter 0]

4 Dec 2019

please see attached response to reviewers

---

## [Decision Letter · Decision Letter 1]

27 Dec 2019

Effect of Voicing and Articulation Manner on Aerosol Particle Emission during Human Speech

PONE-D-19-20208R1

Dear Dr. Ristenpart,

We are pleased to inform you that your manuscript has been judged scientifically suitable for publication and will be formally accepted for publication once it complies with all outstanding technical requirements.

Best regards,

Hui-Ling Yen

Academic Editor

PLOS ONE

Reviewers' comments:

Reviewer's Responses to Questions

**Comments to the Author**

1. If the authors have adequately addressed your comments raised in a previous round of review and you feel that this manuscript is now acceptable for publication, you may indicate that here to bypass the “Comments to the Author” section, enter your conflict of interest statement in the “Confidential to Editor” section, and submit your "Accept" recommendation.

Reviewer #1: All comments have been addressed

Reviewer #2: All comments have been addressed

Reviewer #3: All comments have been addressed

2. Is the manuscript technically sound, and do the data support the conclusions?

Reviewer #1: Yes

Reviewer #2: Yes

Reviewer #3: (No Response)

3. Has the statistical analysis been performed appropriately and rigorously? 

Reviewer #1: Yes

Reviewer #2: Yes

Reviewer #3: (No Response)

4. Have the authors made all data underlying the findings in their manuscript fully available?

Reviewer #1: Yes

Reviewer #2: Yes

Reviewer #3: (No Response)

5. Is the manuscript presented in an intelligible fashion and written in standard English?

Reviewer #1: Yes

Reviewer #2: Yes

Reviewer #3: (No Response)

6. Review Comments to the Author

Reviewer #1: (No Response)

Reviewer #2: (No Response)

Reviewer #3: (No Response)

7. PLOS authors have the option to publish the peer review history of their article (what does this mean?). If published, this will include your full peer review and any attached files.

Reviewer #1: No

Reviewer #2: No

Reviewer #3: No

---

## [Editor Report · Acceptance letter]

7 Jan 2020

PONE-D-19-20208R1 

Effect of Voicing and Articulation Manner on Aerosol Particle Emission during Human Speech 

Dear Dr. Ristenpart:

I am pleased to inform you that your manuscript has been deemed suitable for publication in PLOS ONE. Congratulations! Your manuscript is now with our production department. 

With kind regards,

on behalf of

Dr. Hui-Ling Yen 

Academic Editor

PLOS ONE